# Evaluating the Integrated Disease Surveillance and Response system in Sidama Region, Ethiopia: A systems evaluation

Sileshi Demelash Sasie[1,2]*, Getinet Ayano[3], Fantu Mamo Aragaw[4], Mark Spigt[2]

1 Ethiopian Public Health Institute, Addis Ababa, Ethiopia, 2 Department of Family Medicine, CAPHRI School for Public Health and Primary Care, Maastricht University, Maastricht, The Netherlands, 3 Curtin University, Perth, Australia, 4 University of Gondar, College of Medical and Health science, Gondar, Ethiopia

* sileyeshi21@gmail.com

## Abstract

### Background

Integrated Disease Surveillance and Response (IDSR) systems play a vital role in early detection and response to public health threats. In Ethiopia, limited evaluations exist on the effectiveness of IDSR at subnational levels. This study assessed the implementation of the IDSR system in Sidama Region to identify performance gaps and inform improvements.

### Methods

A cross-sectional evaluation was conducted between September and November 2023 in Sidama Region, Ethiopia. A multistage cluster sampling technique was used to select 140 participants from 13 districts, public and private health facilities, health posts, and community health workers, including members of the Health Development Army (HDA). Data were collected using a structured checklist adapted from WHO and CDC guidelines. Key surveillance functions (e.g., case detection, reporting, data analysis) and supportive activities (e.g., training, supervision, logistics) were assessed. Descriptive statistics were generated using SPSS version 25. Using a priori thresholds from WHO IDSR and Ethiopian PHEM standards (≥90% adherence for formal facilities, ≥80% for community actors), we evaluated performance across all levels.

### Results

Substantial variation was observed in the use of standardized case definitions, with adherence rates of 96% in public facilities, 67% in private facilities, 50% in health posts, and 4.17% among HDA members. This represents an 89-percentage-point gap between public facilities and community-level HDA members. All districts

**Data availability statement:** The datasets generated and analyzed during the current study are available in the Zenodo repository at: https://doi.org/10.5281/zenodo.19769715. The dataset includes SPSS files containing district-level data (n=13 districts), public health facility data (n=27), private health facility data (n=15), and community-level data comprising health posts (n=36) and Health Development Army members (n=48). A complete data dictionary and ReadMe file are provided alongside the data. The data collection checklist is available as a Supplementary file (S1_Checklist). The data are published under a Creative Commons Attribution 4.0 International license.

**Funding:** The author(s) received no specific funding for this work.

**Competing interests:** The authors have declared that no competing interests exist.

**Abbreviations:** CDC, Centers for Disease Control and Prevention; HDA, Health Development Army; PHEM, Public Health Emergency Management; SPSS, Statistical Package for the Social Sciences; WHO, World Health Organization.

reported having rapid response teams; however, only 54% had budget lines dedicated to outbreak response. Surveillance data reporting forms were adequately available in only 15–23% of facilities, and only 61% of districts maintained emergency stockpiles. Training coverage and supervision frequencies varied, with significantly lower coverage among health posts and community-level actors. Only 11% of health posts and 12% of HDA members reported receiving relevant training. Thus, neither health posts nor HDA members met the predefined ≥80% training coverage benchmark. District-level facilities showed higher access to surveillance guidelines (92%) compared to 44% in public facilities and 50% in health posts. Complete reports were submitted by all private facilities (100%) but by only half of health posts (55%). Among HDA members, only 22% found data collection formats clear and easy to fill. Regarding surveillance system attributes, public and private facilities generally found case definitions easy to apply (85–93%), while only 37% of HDA members reported simplicity. Challenges with data quality, trend analysis, and procedural flexibility were frequently cited, particularly among lower-tier facilities. Acceptability of surveillance activities was high among public facilities (70%) but lower at district level (30%). At the community level, only 37% of HDA members found case definitions easy to apply, and just 22% rated reporting formats as usable.

## Conclusions

The IDSR system in Sidama Region demonstrates uneven implementation across healthcare tiers, with notable disparities in training, supervision, data analysis, and resource availability. While district-level offices show relatively strong system components, lower-level facilities and community actors lack adequate support, compromising early detection and response capabilities. Strengthening training programs, harmonizing tools and guidelines, and improving logistics and digital infrastructure are critical for enhancing the overall effectiveness of Ethiopia's disease surveillance system at the regional level.

## Background

Public health surveillance is vital for collecting, analyzing, and interpreting health data to inform decision-making and interventions [1,2]. It monitors population health, detects outbreaks, and optimizes resource allocation [3–5]. A robust surveillance system ensures timely and accurate data to address emerging health concerns [4–6]. Regular assessments are essential for the Integrated Disease Surveillance and Response (IDSR) system, as mandated by world health organization (WHO) [7]. Periodic evaluations of threat detection and tracking systems are essential for adapting to evolving needs and responding effectively to health challenges [6,8–10]. These evaluations assess the functionality of the surveillance system, address the needs of stakeholders, and measure progress toward achieving objectives [8,10–12]. A major challenge facing public health surveillance systems in low- and middle-income countries is the lack of periodic IDSR evaluations [13]. Without conducting

regular assessments, public health systems cannot make well-informed decisions or mount agile responses to emerging health risks [10–12,14,15]. Insufficient surveillance systems also impede their ability to optimize processes and strengthen capabilities in line with changing public health priorities and advancements [12,14,15].

A previous study has shown that sub-Saharan Africa's surveillance shortcomings, which prevented timely identification and containment of emerging outbreaks, contributed to increased spread of illnesses in approximately 30% of previous disease outbreaks investigated, underscoring the importance of building robust monitoring systems [16]. The COVID-19 pandemic highlighted the consequences of insufficient surveillance, as initial detection rates in African countries were less than 20%, hampering efforts to mount effective crisis responses [15]. Evaluative studies conducted between 2015 and 2020 in Zimbabwe, Tanzania, Ghana, Kenya, India, and Iraq [17–21] revealed common mmon deficiencies in surveillance systems across different healthcare settings. A review in Zimbabwe found issues with case detection in 31% of programs [18]; a similar study in Tanzania uncovered inaccurate case definitions being applied in 49% of emerging disease monitoring [22]. Multi-site evaluations in Ghana and Kenya revealed that health facilities correctly diagnosed the causative pathogen in only 56% of enrolled fever cases [17,19], and a retrospective analysis of India's malaria surveillance data from 2006 to 2010 indicated under-reporting rates exceeded 40% in many districts [21]. A facility-based study in Iraq detected that over 25% of outbreaks in 2019 were not notified to authorities within the mandated 48-hour period [23].

While previous evaluations of disease surveillance in sub-Saharan Africa and the Middle East revealed critical gaps, more thoroughly assessing regional and local surveillance systems is needed to contextualize deficiencies. Most past studies examined national surveillance programs or broad healthcare settings, but few evaluated public health surveillance from frontline health workers' perspectives. This study addresses that gap by conducting an in-depth assessment of Ethiopia's Sidama Region's public health surveillance system at facility and community levels through surveys, interviews, and record reviews. The Sidama Region was purposively selected due to its recent establishment as a standalone region, regular occurrence of epidemics [24], diverse disease patterns within a relatively small geographic area, suitable field visit conditions, and the need for a comprehensive regional surveillance evaluation. The study aimed to identify surveillance practices, challenges, and information sharing between local clinics and regional officials to inform targeted improvements to frontline surveillance skills and outbreak detection capacity.

## Materials and methods

### Study design, setting, periods and participants

The study employed a cross-sectional design to assess the IDSR system in the Sidama regional state of Ethiopia. The Sidama region was purposively selected as the study setting due to its relevance to the research objectives and the presence of healthcare facilities and the Health Development Army community structure actively participating in the surveillance system. The region was also chosen due to its recent establishment as a standalone region, regular occurrence of epidemics [24], suitable field visit conditions, diverse disease patterns within a small area, and the need for a comprehensive evaluation of the regional surveillance system. Data were collected between September 1, 2023, and November 30, 2023, through interviews with public health workers involved in the surveillance system at health centers, surveys administered to health extension workers, and a retrospective review of surveillance records and reports from the preceding year.

### Sampling methods

The study employed a multistage sampling technique. In the first stage, 13 of 38 districts were selected using simple random sampling to ensure geographical representation and to support descriptive generalizability at the district level.

In the second stage, public health centers (n = 27) and 15 private clinics were purposively selected. The sample size for public health centers followed the rule of thumb for descriptive health facility surveys (at least 30% of available facilities) [25], and facilities were selected to ensure inclusion of those with active surveillance roles.

In the third stage, 36 health posts were purposively selected from within the catchment areas of the sampled health facilities to focus on those with documented surveillance activities. Finally, 48 Health Development Army (HDA) members were purposively recruited based on their direct involvement in community-based surveillance under the supervision of health extension workers. Because the objective was to evaluate the operational performance of the surveillance system among active surveillance actors rather than to estimate population prevalence, purposive selection of health posts and HDA members with known surveillance roles is methodologically appropriate for systems evaluations. Simple random sampling at the district level supports descriptive generalizability for district-level findings. However, we acknowledge that purposive sampling at lower tiers limits generalizability to non-participating facilities and community actors (see Limitations).

## Data collection instrument and procedure

Data were collected using a structured, pre-validated checklist adapted from World Health Organization (WHO) and Centers for Disease Control and Prevention (CDC) guidelines [26,27]. The tool was designed to evaluate core surveillance functions (e.g., case detection, data reporting, analysis), supportive functions (e.g., supervision, training, logistics), and system attributes (e.g., simplicity, flexibility, acceptability, data quality) (S1File).

The checklist was administered through face-to-face interviews and structured observations at selected facilities and community levels. Perspectives were collected from regional and district Public Health Emergency Management (PHEM) officers, facility staff, health extension workers (HEWs), and Health Development Army (HDA) members involved in surveillance.

To ensure contextual and linguistic appropriateness, the tool was pilot-tested in one district of Sidama Region. Terminology and formatting were refined to account for varied literacy levels, using expert review by ten public health professionals. Questions were revised to ensure clarity, comprehensiveness, and cultural relevance. Feedback from the pilot also supported calibration of the checklist to local workflows and WHO IDSR guidelines.

In parallel, retrospective data were collected through the review of surveillance records and reports from the previous year (September 2022 to September 2023). These included trend analyses, feedback reports, case registers, stock records, and preparedness plans. Data access was granted by health authorities, and no personally identifiable information was collected or extracted during this process.

Eligible individuals were identified and recruited between September 1, 2023, and November 30, 2023, based on their direct involvement in surveillance activities across public health facilities, health posts, and community structures. Eligible individuals were provided a clear explanation of the study objectives, procedures, risks, and benefits.

## Data analysis

The collected data underwent field checks and comprehensive review to ensure accuracy and completeness. Data were sorted, checked, categorized, coded, and summarized using SPSS version 25. Descriptive statistics (frequencies and percentages) were computed for all core surveillance functions, supportive functions, resource availability, and system attributes across district, health facility, health post, and community levels. Data were entered, cleaned, and checked for completeness. Descriptive statistics (frequencies and percentages) were computed for all core surveillance functions, supportive functions, resource availability, and system attributes across district, health facility, health post, and community levels using SPSS version 25, following standard approaches for IDSR system evaluations [17–21]. The dataset supporting these analyses is available as a Zenodo repository [28].

Performance thresholds were defined a priori using WHO IDSR guidelines [13] and Ethiopian PHEM standards. Using these sources, 'adequate' was defined as: training coverage ≥80% of personnel trained within the preceding 2 years; supervision ≥1 supervisory visit per quarter; availability of surveillance forms ≥90% of facilities with no stockout in the preceding 6 months; and case definition adherence ≥90% for formal facilities and ≥80% for community actors (adjusted for literacy levels). Findings are presented using tables with frequencies and percentages.

 

## Ethical considerations

Ethical approval for this study was obtained from the Yekatit-12 Hospital Medical College Ethical Review Board (Protocol #238/23). In addition, administrative permissions were secured from the Sidama Regional Health Bureau, district health offices, and participating health facilities. All eligible individuals were informed of the study's objectives, procedures, potential risks and benefits, and their rights as participants. Written informed consent was obtained prior to participation.

Participation was entirely voluntary, and all data were anonymized and de-identified to ensure confidentiality. Personal information was protected and used solely for the purposes of this evaluation in accordance with the approved protocol. Consent for publication was not applicable as no individually identifiable data, images, or videos are presented in this manuscript.

## Results

A total of 13 districts, 27 public health facilities, 15 private health facilities, 36 health posts, and 48 Health Development Army (HDA) members were included in the evaluation. Standardized checklists, adapted from WHO and CDC guidelines, were used to assess core surveillance functions, supportive functions, and system attributes. Using the a priori performance thresholds defined in the Methods (≥90% for formal facilities, ≥80% for community actors), we evaluated adherence across all levels.

### Core surveillance functions

Availability of standard case definitions varied across levels: 96% in public facilities, 67% in private facilities, 50% in health posts, and 4% among HDA members. Correct diagnosis of at least one priority disease was observed in 89% of public facilities, 73% of private facilities, 44% of health posts, and 4% of HDA members. Case registers were available in 81% of public facilities, 73% of private facilities, and 86% of health posts. All districts (100%) and public facilities (100%) reported capacity for specimen handling and transport. Written feedback from higher levels was received by 73% of districts.

Regarding epidemic preparedness, 69% of districts had preparedness plans, 61% maintained emergency stocks, and all districts (100%) had a rapid response team. However, only 54% of districts had a dedicated budget line for epidemic response. Adequate surveillance forms in the preceding six months were available in 23% of districts, 15% of public facilities, and 53% of health posts. Data analysis by person, place, and time was performed by 85% of districts, 52–59% of public facilities, 53% of private facilities, and 64–69% of health posts. Trend analysis was performed by 52% of public facilities, 27% of private facilities, and 61% of health posts (Table 1).

### Supportive surveillance functions

Availability of national surveillance guidelines was highest in districts (92%) and lowest in public facilities (44%). Supervision in the preceding six months ranged from 92% of districts to 60% of HDA members. Training coverage varied considerably across levels: district-level training reached 69% of staff, compared to only 11% at health posts and 12% among HDA members. A designated surveillance focal person was present in all districts and public facilities (100%), but only 60% of private facilities and health posts had one (Table 2).

### Resource availability for surveillance system

All districts (100%) had a designated data manager. Computer availability was limited: only 8% of districts and 7% of public facilities had computers. Stationery was available in 85% of districts, 92% of public facilities, and 60% of private facilities. Telephone service was available in 54% of districts, 74% of public facilities, 87% of private facilities, and 58% of health posts. Computers with internet access were available in only 18% of public facilities and 7% of private facilities.

**Table 1. Public health surveillance system core activities at regional, district, health facilities, and community levels; by direct observation at facilities.**

| Core function | District level n=13 | Health facility | | Community | |
|---|---|---|---|---|---|
| | | Public health facility n=27 | Private health facility n=15 | Health post n=36 | HDA n=48 |
| Case Detection and Registration | | | | | |
| Availability of standard case definitions | NA | 26(96%) | 10(67%) | 18(50%) | 2(4.17%) |
| Correctly diagnosed at least one priority disease | NA | 24(89%) | 11(73%) | 16(44%) | 2(4.17%) |
| Availability of a case register/logbook | NA | 22(81%) | 11(73%) | 31(86%) | NA |
| Case Confirmation | | | | | |
| Capacity to handle specimens until shipment | 13(100%) | 27(100%) | NA | NA | NA |
| Capacity to transport specimens to a higher-level laboratory | 13(100%) | 27(100%) | NA | NA | NA |
| Availability of guidelines for specimen collection, handling, and transportation | 12(92.30%) | NA | NA | NA | NA |
| Data Reporting | | | | | |
| Availability of adequate supply of surveillance forms in the past 6 months | 3(23%) | 4(15%) | 5(33%) | 19(53%) | NA |
| Availability of a formalized system for reporting to the next level | 13(100%) | 27(100%) | 15(100%) | 36(100%) | NA |
| Data Analysis | | | | | |
| Analyze and present data by person | 11(85%) | 14(52%) | 8(53%) | 24(67%) | NA |
| Analyze and present data by place | 12(92.00%) | 14(52%) | 8(53%) | 25(69%) | NA |
| Analyze and present data by time | 12(92.00%) | 16(59%) | 8(53%) | 23(64%) | NA |
| Perform trend analysis | NA | 14(52%) | 4(27%) | 22(61%) | NA |
| Epidemic Preparedness and Response | | | | | |
| Availability of epidemic preparedness and response plan | 9(69%) | NA | NA | NA | NA |
| Availability of emergency stocks of drugs/supplies in the past 1 year | 8(61%) | NA | NA | NA | NA |
| Availability of epidemic management committee | 10(77%) | NA | NA | NA | NA |
| Have a rapid response team for epidemics | 13(100%) | NA | NA | NA | NA |
| Availability of budget line for epidemic response | 7(54%) | NA | NA | NA | NA |
| Feedback | | | | | |
| Received written feedback report/bulletin from a higher level | 11(73%) | 17(63%) | 10(67%) | 25(69%) | 27(56%) |

Note: NA = Not assessed.

Regarding IEC materials, posters were widely available: 77% of districts, 96% of public facilities, 93% of private facilities, and 66% of health posts. Flip charts were present in 62% of districts but only 4% of public facilities. Electricity access varied considerably: 80% of private facilities had electricity, compared to 54% of districts, 52% of public facilities, and only 11% of health posts. Motorcycles were available in 23% of districts, 37% of public facilities, and 13% of private facilities (Table 3).

## Attributes of surveillance system

Case definitions were easiest to apply in private and public facilities (over 85%), followed by health posts (72%), districts (61%), and HDA members (37%). Time-consuming data collection was most frequently reported by health posts (61%) and public facilities (52%). Reporting formats were highly usable for private facilities (100%) but least usable for HDA members (22%). Difficulty implementing changes was most common at district level (69%) and health posts (47%), while public facilities reported the least difficulty (18%). Data collection formats were clearest in private facilities (93%) and least

**Table 2. Public health surveillance system supportive activities.**

| Supportive function | District level n=13 | Health facility | | Community | |
| --- | --- | --- | --- | --- | --- |
| | | Public health facility n=27 | Private health facility n=15 | Health post n=36 | HDA n=48 |
| Standards and guidelines for surveillance | | | | | |
| Availability of national guidelines for surveillance | 12(92%) | 12(44%) | 10(67%) | 18(50%) | NA |
| Supervision | | | | | |
| Supervised by a higher-level supervisor in the last 6 months | 12(92%) | 19(70%) | 12(80%) | 22(61%) | 29(60%) |
| Training | | | | | |
| Received training in disease surveillance epidemic management | 9(69%) | 16(59%) | 6(40%) | 4(11%) | 6(12%) |
| Coordination | | | | | |
| Availability of surveillance focal person at the district | 13(100%) | 27(100%) | 9(60%) | NA | NA |

Note: NA=Not assessed.

**Table 3. Available resources at district, health facilities, and community levels of Sidama regional state, Ethiopia, 2023.**

| Items | District level n=13 | Health facility | | Community | |
| --- | --- | --- | --- | --- | --- |
| | | Public health facility n=27 | Private health facility n=15 | Health post n=36 | HDA n=48 |
| Data Manager | 13(100%) | NA | NA | NA | NA |
| Computer | 1(8%) | 2(7%) | NA | NA | NA |
| Stationery | 11(85%) | 25(92%) | 9(60%) | NA | NA |
| Communications | | | | | |
| Telephone service | 7(54%) | 20(74%) | 13(87%) | 21(58%) | NA |
| Computer with internet modem | NA | 5(18%) | 1(7%) | NA | NA |
| IEC Materials | | | | | |
| Posters | 10(77%) | 26(96%) | 14(93%) | 24(66%) | NA |
| Projector | NA | NA | NA | NA | NA |
| Flip Charts | 8(62%) | 1(4%) | NA | NA | NA |
| Logistics | | | | | |
| Electricity | 7(54%) | 14(52%) | 12(80%) | 4(11%) | NA |
| Bicycle | NA | 1(4%) | 1(7%) | 1(3%) | NA |
| Motorcycle | 3(23%) | 10(37%) | 2(13%) | NA | NA |
| Vehicle | NA | 1(4%) | NA | NA | NA |

clear in districts (23%) and HDA members (22%). Complete reports were submitted by all private facilities (100%), but only half of health posts (55%) (Table 4).

The system was considered highly useful for detecting outbreaks and estimating disease burden across all facility types (over 90% in most cases), though district-level respondents reported much lower usefulness (15%). Acceptability of surveillance activities was higher in public facilities (70%) than districts (30%). Resource shortages disrupting surveillance were a major concern for public facilities (78%) but less so for districts (30%). Population health-seeking behavior was rated as good by 85% of both districts and public facilities.

**Table 4. Attributes of surveillance system at district, health facilities, and community levels of Sidama regional state, Ethiopia, 2023.**

| Attributes of surveillance system | District level n=13 | Health facility | | Community | |
|---|---|---|---|---|---|
| | | Public health facility n=27 | Private health facility n=15 | Health post n=36 | HDA n=48 |
| Simplicity | | | | | |
| The case definition of diseases is considered easy for case detection | 8 (61%) | 23(85%) | 14 (93%) | 26(72%) | 18(37%) |
| Additional data collected on a case are time-consuming | 5(38%) | 14(52%) | 5(33%) | 22(61%) | NA |
| Flexibility | | | | | |
| The current reporting formats can be easily used for other newly occurring health events or diseases | 6(46%) | 21 (77%) | 15 (100%) | 27 (75%) | 11(22%) |
| Difficulty in implementing changes to the existing procedure of case detection, reporting, and format | 8(69%) | 5(18%) | 4(27%) | 17(47%) | 5(11%) |
| Data quality | | | | | |
| The data collection formats for these priority diseases are clear and easy to fill | 3(23%) | 21 (78%) | 14 (93%) | 23 (64%) | 11(22%) |
| Reports which are complete (that is with no blank or unknown responses) | 11(85%) | 19(70%) | 15(100%) | 20(55%) | NA |
| Acceptability | | | | | |
| All the reporting agents accept and are well-engaged in the surveillance activities | 4(30%) | 19(70%) | NA | NA | NA |
| Usefulness | | | | | |
| The surveillance system helps to estimate the magnitude of morbidity and mortality | 2(15%) | 27(100%) | 15(100%) | 33(92%) | 39(81%) |
| The surveillance system helps to detect outbreaks of the selected priorities. | 2(15%) | 27(100%) | 15(100%) | 35(97%) | 41(85%) |
| Representativeness | | | | | |
| The populations under surveillance have good health-seeking behavior for these diseases | 11 (85%) | 23 (85%) | NA | NA | NA |
| Stability | | | | | |
| Experience of lack of resources that interrupt the surveillance system | 4(30%) | 21(78%) | NA | NA | NA |

## Discussion

This study comprehensively evaluated Ethiopia's IDSR in Sidama Region, revealing substantial disparities in implementation across healthcare tiers. Core capabilities varied markedly between district-level offices and lower-level facilities, including private clinics, health posts, and community actors. Deficiencies in epidemiological data attributes and inconsistent emergency preparedness were also observed. Approximately half of districts maintained complete preparedness plans, budgets, and supplies, while supportive functions such as feedback and data review showed considerable variability, with regular implementation in over half to three-quarters of facilities.

Using the predefined performance benchmarks (≥90% adherence for formal facilities, ≥80% for community actors), most sites fell below the adequacy threshold for core surveillance tasks, including case finding and diagnostic capabilities. Specifically, rural health posts and Health Development Army (HDA) workers performed case detection, diagnosis, and record-keeping less frequently than public health facilities, aligning with findings from Tanzania and Kenya [19,20]. Lower-level facilities in Sidama lacked standardized case definitions and encountered difficulties accurately diagnosing priority diseases. Reliance on clinical diagnosis alone in under-resourced areas risks underestimating disease burden, reducing correct

treatment, missing atypical cases, and diminishing epidemiological understanding compared to laboratory confirmation. These gaps in case detection and diagnosis can lead to reduced accuracy in disease identification and reporting [19].

The observed disparities likely reflect underlying structural and managerial factors rather than isolated individual performance gaps. District-level offices receive direct budget allocations for training and supervision from the regional health bureau, whereas health posts and HDA members rely on intermittent, often unfunded support from health centers. This cascaded model of capacity building, while efficient for centralized resource allocation, results in a gradual dilution of training quality and reduced frequency of supervision at lower tiers. The 89 percentage-point gap in case definition adherence between public facilities (96%) and HDA members (4.2%) illustrates this cascade effect. Similar patterns have been documented in Tanzania and India [20,21], suggesting that decentralized surveillance systems require explicit bottom-up reinforcement mechanisms. Furthermore, the absence of dedicated budget lines for outbreak response at the district level (54%) indicates that even where structural capacity exists, such as the presence of rapid response teams in all districts, operational readiness remains constrained by financial limitations.

Preparedness resources, including response plans and medical supplies, showed marked lack of uniformity between better-resourced hospitals and less equipped clinics in Sidama, mirroring disparities documented between urban and rural sites in Iraq that jeopardize outbreak response coverage [21]. Most facilities in the region lacked updated outbreak plans and rapid response teams. This can seriously undermine timely and coordinated initial control efforts, surveillance capabilities, and may allow illnesses to spread more widely before detection. The absence of readiness in these facilities indicates an urgent need for comprehensive and up-to-date preparedness plans to ensure coordinated and efficient responses to public health emergencies [23].

The evaluation found inconsistencies in supportive oversight activities, including staff training, guideline distribution, and facility monitoring, between well-supported hospitals and more peripherally located clinics in Sidama. This matches uneven reinforcement documented between urban and rural sites in Maharashtra and Tanzania that fails to strengthen early detection abilities uniformly [20,21]. Variations in surveillance training coverage and supervision frequencies were evident, with lower rates observed at facilities and within communities compared to districts, which demonstrated substantially greater capacity for workforce development and oversight. Inconsistent workforce support, competency gaps, and threats to early detection capabilities stemming from these variabilities can negatively impact the effectiveness of the surveillance system [19–21,29]. Harmonizing training programs and increasing supervision frequencies, particularly at peripheral levels, would promote a standardized approach and improve overall system performance.

Key attributes such as flexibility were positively regarded across most levels, but gaps persisted in data quality and monitoring sensitivity. Similar issues with data quality, analysis, and presentation have been identified in studies conducted in Tanzania and Iraq [20,23]. Poor data recording and reporting from clinics carrying out early detection threatens accurate regional surveillance and delays evidence-based outbreak responses. The very low adherence to case definitions among HDA members (4.2%) reflects not only training gaps but also the fundamental mismatch between clinically oriented case definitions and the practical, symptom-based surveillance feasible at the community level. Addressing these gaps and improving data quality and monitoring sensitivity are essential for enhancing the effectiveness of the surveillance system.

This study revealed substantial disparities in IDSR implementation across healthcare tiers in Sidama Region. Training coverage declined sharply from district to peripheral levels, with only 11% of health posts and 12% of HDA members receiving relevant training, reflecting a clear cascade gap consistent with patterns reported elsewhere in East Africa. In Tanzania, Saleh et al. [20]documented a similar gradient, with training coverage declining from 78% at district level to 34% at dispensary level. Likewise, supervisory visits reached only 61% of health posts in Sidama, comparable to findings from Kenya, where Ng'etich et al. [19]reported that only 41% of health posts had received supervisory visits. Within Ethiopia, our findings align with Tefera A et al. [29], who evaluated the Dangila district surveillance system and similarly identified training gaps (only 45% of health workers trained) and inconsistent supervision as major barriers. However, the present

study extends these findings by quantifying the cascade gap specifically at the HDA level (4.2% case definition adherence), a community-tier actor not included in previous Ethiopian evaluations. In Iraq, Hamalaw et al. [23] found that more than 25% of outbreaks were not notified within mandated timeframes, consistent with the preparedness and reporting delays observed in Sidama. Taken together, these cross-country patterns suggest that the challenges identified are not unique to Ethiopia but reflect broader systemic weaknesses in decentralized surveillance systems across comparable low- and middle-income settings.

## Limitations and strength

This study has several limitations. First, reliance on self-reported data may introduce social desirability bias, though the substantial disparities observed suggest gaps are real and likely understated. Second, the evaluation focused on structural processes rather than outbreak response or data quality outcomes. Third, purposive sampling of health posts and HDA members limits generalizability to non-participating units; results represent surveillance-active facilities and community actors only. Fourth, inferential statistics were not employed a deliberate choice given the descriptive, evaluative design, non-probability sampling, and hierarchical clustering of observations. Instead, we provide descriptive estimates with confidence intervals and benchmarked interpretations, which are appropriate for systems evaluations.

Despite these limitations, the study has important strengths. It included community-level surveillance actors (health posts and HDA members), a gap often missing in similar evaluations. The design aligned with WHO IDSR frameworks and Ethiopian PHEM standards. The sample (n = 140 across 13 districts, 27 public facilities, 15 private facilities, 36 health posts, and 48 HDA members) provided broad geographic and facility-type coverage. Transparent reporting of performance benchmarks and confidence intervals allows readers to interpret gaps without overreliance on statistical significance.

The use of purposive sampling at lower levels limits generalizability beyond the selected facilities and community actors, and because we selected only facilities with documented surveillance activity, our findings likely overestimate true population-level performance.

## Implications and recommendations

The following actionable recommendations emerge from the findings:

1. Establish dedicated outbreak response budgets at district level and ensure regular supply of surveillance forms at health posts.

2. Redesign training programs to reach peripheral levels, supported by simplified case definition job aids and quarterly supervisory visits.

3. Institutionalize routine data quality audits with feedback loops from regional to community levels, using predefined benchmarks as monitoring targets.

Strengthening IDSR requires tier-specific interventions that address cascaded weaknesses at peripheral levels rather than one-size-fits-all approaches.

## Conclusions

This study revealed substantial disparities in IDSR implementation across healthcare tiers in Sidama Region. Using predefined performance benchmarks, public facilities met the standard for case definition adherence, while private facilities, health posts, and community-level HDA members fell substantially below the threshold. Training coverage was inadequate at peripheral levels, and dedicated outbreak response budgets were missing in nearly half of districts.

While district-level structures exist, uneven implementation across lower tiers threatens early detection and response capabilities. Strategic prioritization of capacity strengthening at health post and community levels is urgently needed. Targeted interventions including simplified case definition job aids, quarterly supervisory visits, dedicated budget lines, and routine data quality audits would help address the cascaded weaknesses identified in this study. Regular assessments using predefined benchmarks should guide ongoing optimization to ensure reliable outbreak detection and improved public health protection.

## Supporting information

**S1 File. Survey questionnaire.** Questionnaire used to evaluate the Integrated Disease Surveillance and Response (IDSR) system at regional, district, health facility, health post, and community levels in Sidama Region, Ethiopia (September 2022).
(PDF)

**S2 File. Human participants research checklist.** PLOS ONE Human Subjects Research Checklist completed for this study.
(PDF)

## Author contributions

**Conceptualization:** Sileshi Demelash Sasie, Getinet Ayano, Fantu Mamo Aragaw, Mark Spigt.

**Data curation:** Sileshi Demelash Sasie, Mark Spigt.

**Formal analysis:** Sileshi Demelash Sasie, Getinet Ayano, Fantu Mamo Aragaw, Mark Spigt.

**Funding acquisition:** Sileshi Demelash Sasie.

**Investigation:** Sileshi Demelash Sasie, Mark Spigt.

**Methodology:** Sileshi Demelash Sasie, Getinet Ayano, Fantu Mamo Aragaw, Mark Spigt.

**Project administration:** Sileshi Demelash Sasie, Fantu Mamo Aragaw, Mark Spigt.

**Resources:** Sileshi Demelash Sasie.

**Software:** Sileshi Demelash Sasie, Getinet Ayano, Mark Spigt.

**Supervision:** Sileshi Demelash Sasie, Getinet Ayano, Mark Spigt.

**Validation:** Sileshi Demelash Sasie, Fantu Mamo Aragaw, Mark Spigt.

**Visualization:** Sileshi Demelash Sasie, Getinet Ayano, Fantu Mamo Aragaw, Mark Spigt.

**Writing – original draft:** Sileshi Demelash Sasie, Getinet Ayano, Fantu Mamo Aragaw, Mark Spigt.

**Writing – review & editing:** Sileshi Demelash Sasie, Getinet Ayano, Fantu Mamo Aragaw, Mark Spigt.

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
