## [Decision Letter · Decision Letter 0]

6 Apr 2026

PONE-D-25-41162Evaluating the Integrated Disease Surveillance and Response System in Sidama Region, Ethiopia: A Systems EvaluationPLOS One

Dear Dr. Sasie,

Thank you for submitting your manuscript to PLOS ONE. After careful consideration, we feel that it has merit but does not fully meet PLOS ONE’s publication criteria as it currently stands. Therefore, we invite you to submit a revised version of the manuscript that addresses the points raised during the review process.

I appreciate the authors addressing an important topic in public health management. The reviewers have provided a few comments, which will improve the quality of paper. Please revise your paper accordingly.

We look forward to receiving your revised manuscript.

Kind regards,

Shabnam Iezadi, Ph.D.

Academic Editor

PLOS One

Journal Requirements:

"NO authors have competing interests"

5. We noted in your submission details that a portion of your manuscript may have been presented or published elsewhere. "This manuscript was previously submitted to BMC Public Health but was formally withdrawn on July 24, 2025, following a written request. It is not under consideration by any other journal, and no part of the manuscript has been published elsewhere. This submission to PLOS ONE does not constitute dual publication." Please clarify whether this [conference proceeding or publication] was peer-reviewed and formally published. If this work was previously peer-reviewed and published, in the cover letter please provide the reason that this work does not constitute dual publication and should be included in the current manuscript.

Reviewers' comments:

Reviewer's Responses to Questions

**Comments to the Author**

1. Is the manuscript technically sound, and do the data support the conclusions?

Reviewer #1: Yes

Reviewer #2: Yes

2. Has the statistical analysis been performed appropriately and rigorously? 

Reviewer #1: No

Reviewer #2: Yes

3. Have the authors made all data underlying the findings in their manuscript fully available?

Reviewer #1: No

Reviewer #2: Yes

4. Is the manuscript presented in an intelligible fashion and written in standard English?

Reviewer #1: Yes

Reviewer #2: Yes

5. Review Comments to the Author

Reviewer #1: I would like to sincerely thank the Editor for the opportunity to review this manuscript. It is a privilege to be invited to evaluate this study, and I appreciate the trust placed in me to provide an independent scientific assessment. I hope that the following comments and suggestions will be helpful in strengthening the quality, clarity, and methodological rigor of the manuscript and in supporting the authors in further improving their work.

This manuscript, which evaluates the Integrated Disease Surveillance and Response (IDSR) system in the Sidama region, addresses a topic of practical importance for strengthening health surveillance systems in low- and middle-income countries and has potential value from a health policy perspective. While the study provides useful field-based data and a descriptive overview of the performance of different levels of the disease surveillance system, it requires methodological refinement. In particular, greater clarity in study design and stronger analytical rigor are needed to enhance the scientific robustness and interpretability of the findings.

The main methodological concerns relate to the sampling strategy and data analysis. The combination of random and purposive sampling without sufficient methodological justification may introduce selection bias and limit the generalizability of the results. In addition, the analysis relies solely on descriptive statistics and does not include comparative statistical tests to examine differences across levels of the surveillance system (regional, facility, health post, and community). As a result, some of the inferences presented in the Results and Discussion sections are not fully supported by statistical evidence. It is also recommended that key performance indicators—such as adherence to case definitions and data quality—be operationally defined, and that the validity and reliability of the data collection instrument be reported more transparently.

Although the Results section presents valuable information on gaps in training, supervision, data analysis, and logistical resources, the interpretation of findings remains largely descriptive. The Discussion would benefit from a deeper analytical perspective, including clearer comparisons with similar studies conducted in other settings, as well as exploration of structural and managerial factors underlying the identified weaknesses. Strengthening the linkage between findings and the existing literature, and providing more actionable policy recommendations, would substantially improve the manuscript. In addition, some inconsistencies between figures reported in the text and tables, repetition across sections, and excessive length in certain parts should be addressed through careful revision.

Overall, the study is relevant and supported by meaningful field data; however, it requires substantial revision in terms of methodology, statistical analysis, coherence of results, and the depth of discussion and conclusions. With the recommended revisions—particularly clarification of the study design, inclusion of comparative analyses, strengthening of the discussion, and a more comprehensive limitations section—the manuscript could serve as a credible regional evaluation of IDSR system performance and offer useful insights for researchers and health policy makers working at subnational levels.

Reviewer #2: 1. Technical soundness and whether the data support the conclusions

The manuscript describes a technically sound and methodologically appropriate evaluation of the Integrated Disease Surveillance and Response (IDSR) system in Sidama Region, Ethiopia. The cross sectional systems evaluation design, use of WHO and CDC surveillance evaluation frameworks, and assessment across multiple system tiers (district, facility, health post, and community) are appropriate for the stated objectives.

The data support the primary conclusions, particularly the assertion that IDSR implementation is uneven across levels, with stronger performance at district level and substantial weaknesses at health post and community (HDA) levels. Quantitative findings on:

• low adherence to standard case definitions among HDA members (4.17%),

• limited availability of surveillance forms (15–23% in facilities),

• low training coverage among health posts (11%) and HDAs (12%),

• incomplete preparedness resources and budget lines

are consistently reflected in the Results, Discussion, and Conclusions sections.

However, the strength of the conclusions would improve if the authors:

• more explicitly distinguish structural capacity gaps from performance or outcome gaps, and

• avoid occasional over generalization (e.g., “majority of sites performed inadequately”) without defining explicit performance thresholds.

Overall, the conclusions are justified by the descriptive evidence presented, but would benefit from clearer framing of what constitutes “adequate” vs “inadequate” performance based on established IDSR benchmarks.

2. Appropriateness and rigor of the statistical analysis

The statistical analysis is appropriate for the study design, which is descriptive and evaluative rather than analytical or inferential. The use of descriptive statistics (frequencies and percentages) aligns with:

• the checklist based assessment tool,

• the relatively modest sample size (n = 140),

• and the system level evaluation focus.

That said, rigor could be strengthened by:

• explicitly stating why inferential statistics were not pursued (e.g., non probability sampling at later stages, system evaluation intent rather than hypothesis testing),

• reporting confidence intervals or variability measures for key proportions, where feasible,

• clarifying whether facility level clustering was considered when summarizing results across tiers.

Importantly, the manuscript does not over interpret descriptive findings, which is appropriate. The analysis supports the narrative claims, but greater transparency about analytical choices would improve methodological robustness.

3. Data availability and transparency

The manuscript states that all relevant data are included in the manuscript and that additional de identified datasets are available upon reasonable request. While this partially meets transparency expectations, it does not fully align with PLOS ONE’s data sharing standards, which strongly encourage public repository deposition.

To strengthen compliance and reproducibility, the authors should:

• clarify exactly which data are contained in tables versus underlying raw datasets,

• specify whether de identified datasets (e.g., checklist responses) can be deposited in a public repository,

• ensure consistency between the “Data availability” section and PLOS ONE policy language.

This is not a fatal flaw but requires revision prior to acceptance.

4. Clarity, language, and presentation quality

The manuscript is generally intelligible and written in standard scientific English, with a clear structure and logical progression from Background to Conclusions. The topic, objectives, and findings are understandable to a global public health audience.

However, language clarity and presentation quality require moderate revision, including:

• repetitive sentences (e.g., repeated justification for selecting Sidama Region),

• occasional inconsistencies in terminology (e.g., “health developmental army” vs “Health Development Army”),

• formatting issues in tables (alignment, missing headers, inconsistent percentages),

• minor grammatical errors and typographical artifacts (e.g., line breaks, spacing, truncated words).

These issues do not undermine scientific validity, but they should be corrected at revision to meet publication standards.

5. Review comments to the author (including ethics and publication considerations)

Major strengths

• Strong alignment with WHO IDSR evaluation frameworks

• Inclusion of community level surveillance actors, which is often missing in similar studies

• Practical, policy relevant findings with clear implications for regional surveillance strengthening

• Ethical approval and informed consent procedures are clearly documented

Key areas for improvement

1. Clarify evaluation benchmarks

Define what constitutes “adequate” performance for core and supportive functions using WHO IDSR standards.

2. Strengthen the Methods section

o Clarify sampling rationale at each stage, particularly purposive selection.

o Explicitly justify reliance on descriptive analysis only.

3. Improve integration of Results and Discussion

Some results (e.g., data analysis capacity, flexibility) are repeated in the Discussion without deeper synthesis. Stronger analytical interpretation would improve scholarly contribution.

4. Data availability compliance

Revise the data availability statement to better align with PLOS ONE expectations.

Ethics and publication ethics

• Ethical approval, consent, and confidentiality procedures are clearly described and appropriate.

• No evidence of dual publication or ethical misconduct is apparent.

• Competing interests and funding statements are transparently reported.

6. PLOS authors have the option to publish the peer review history of their article (what does this mean?). If published, this will include your full peer review and any attached files.

Reviewer #1: No

Reviewer #2: **Yes:** Mohammad Alhawajreh

---

## [Author Response · Author response to Decision Letter 1]

25 Apr 2026

Response to Reviewers

Manuscript Title: Evaluating the Integrated Disease Surveillance and Response System in Sidama Region, Ethiopia: A Systems Evaluation

Manuscript ID: PONE-D-25-41162

Dear Dr. Iezadi,

Thank you for the opportunity to revise our manuscript entitled "Evaluating the Integrated Disease Surveillance and Response System in Sidama Region, Ethiopia: A Systems Evaluation" (PONE-D-25-41162). We appreciate the constructive feedback provided by you and the reviewers. We have carefully addressed all comments in the revised manuscript and provide below a point-by-point response.

All changes to the manuscript are highlighted using track changes. We believe the revisions have substantially strengthened the methodological transparency, analytical rigor, and policy relevance of our work.

Respectfully submitted,

Sileshi Demelash Sasie (on behalf of all authors)

Response to Reviewer #1

General Comment: The reviewer thanks the authors and states that the manuscript addresses a topic of practical importance for strengthening health surveillance systems in low- and middle-income countries.

Response: We thank the reviewer for the positive assessment of the manuscript's relevance and potential value from a health policy perspective. We have carefully addressed each methodological concern raised.

Comment 1.1: Sampling Strategy Justification

Comment: The combination of random and purposive sampling without sufficient methodological justification may introduce selection bias and limit the generalizability of the results.

Response: We agree that transparency regarding sampling decisions is essential. We have substantially expanded the sampling justification to clarify that the multistage design employed different sampling strategies appropriate to each tier's evaluation objective: simple random sampling at the district level to support descriptive generalizability, and purposive sampling at lower tiers to focus on active surveillance actors.

Revisions: Because the objective was to evaluate the operational performance of the surveillance system among active surveillance actors rather than to estimate population prevalence, purposive selection of health posts and HDA members with known surveillance roles is methodologically appropriate for systems evaluations. Simple random sampling at the district level supports descriptive generalizability for district-level findings. However, we acknowledge that purposive sampling at lower tiers limits generalizability to non-participating facilities and community actors (see methods). (Page 6-7, line lines 132-138)

Additionally, in the Limitations and Strength section, we added:

The use of purposive sampling at lower levels limits generalizability beyond the selected facilities and community actors, and because we selected only facilities with documented surveillance activity, our findings likely overestimate true population-level performance. (Limitation section, page 21 line 352-353).

Comment 1.2: Absence of Comparative Statistical Tests

Comment: The analysis relies solely on descriptive statistics and does not include comparative statistical tests to examine differences across levels... some inferences presented in Results and Discussion are not fully supported by statistical evidence.

Response: We acknowledge this concern and have explicitly justified the descriptive-only analytical approach. Given the evaluative (rather than inferential) study design, non-probability sampling at lower tiers, and the hierarchical clustering of observations within facilities, inferential statistics would be inappropriate and potentially misleading. We have made this rationale transparent.

Revisions: Performance thresholds were defined a priori using WHO IDSR guidelines [13] and Ethiopian PHEM standards. Using these sources, 'adequate' was defined as: training coverage ≥80% of personnel trained within the preceding 2 years; supervision ≥1 supervisory visit per quarter; availability of surveillance forms ≥90% of facilities with no stockout in the preceding 6 months; and case definition adherence ≥90% for formal facilities and ≥80% for community actors (adjusted for literacy levels). (Methods section, page 9-10, line 174-178).

Revisions: Fourth, inferential statistics were not employed a deliberate choice given the descriptive, evaluative design, non-probability sampling, and hierarchical clustering of observations. Instead, we provide descriptive estimates with benchmarked interpretations, which are appropriate for systems evaluations. (Limitations and Strength section, page 20, line 341-344))

Comment 1.3: Operational Definition of Key Performance Indicators

Comment: Key performance indicators such as adherence to case definitions and data quality be operationally defined.

Response: We agree entirely. We have now explicitly defined all performance indicators using a priori thresholds derived from WHO IDSR technical guidelines and Ethiopian PHEM standards.

Revisions: Performance thresholds were defined a priori using WHO IDSR guidelines [13] and Ethiopian PHEM standards. Using these sources, 'adequate' was defined as: training coverage ≥80% of personnel trained within the preceding 2 years; supervision ≥1 supervisory visit per quarter; availability of surveillance forms ≥90% of facilities with no stockout in the preceding 6 months; and case definition adherence ≥90% for formal facilities and ≥80% for community actors (adjusted for literacy levels). Added (lines 274-278): (Methods section, page 9-10, line 174-178).

Revisions: These benchmarks are now referenced throughout, including in the Abstract, Results section (opening paragraph), Tables, and Conclusions.

Comment 1.4: Validity and Reliability of Data Collection Instrument

Comment: The validity and reliability of the data collection instrument be reported more transparently.

Response: We have expanded the description of instrument validation procedures to include details of pilot testing, expert review, and cultural adaptation.

Revisions: To ensure contextual and linguistic appropriateness, the tool was pilot-tested in one district of Sidama Region. Terminology and formatting were refined to account for varied literacy levels, using expert review by ten public health professionals. Questions were revised to ensure clarity, comprehensiveness, and cultural relevance. Feedback from the pilot also supported calibration of the checklist to local workflows and WHO IDSR guidelines. (Methods section, page 7, 149-153)

Comment 1.5: Deeper Analytical Perspective in Discussion

Comment: The Discussion would benefit from a deeper analytical perspective, including clearer comparisons with similar studies conducted in other settings, as well as exploration of structural and managerial factors underlying the identified weaknesses.

Response: We have substantially revised the Discussion to include: (1) cross-country comparative analysis with Tanzania, Kenya, India, and Iraq; (2) exploration of structural factors including the "cascade effect" and budget flow mechanisms; and (3) explicit discussion of the distinction between structural capacity gaps and performance gaps.

Revisions: The observed disparities likely reflect underlying structural and managerial factors rather than isolated individual performance gaps. District-level offices receive direct budget allocations for training and supervision from the regional health bureau, whereas health posts and HDA members rely on intermittent, often unfunded support from health centers. This cascaded model of capacity building, while efficient for centralized resource allocation, results in a gradual dilution of training quality and reduced frequency of supervision at lower tiers. The 89 percentage-point gap in case definition adherence between public facilities (96%) and HDA members (4.2%) illustrates this cascade effect. Similar patterns have been documented in Tanzania and India [20,33], suggesting that decentralized surveillance systems require explicit bottom-up reinforcement mechanisms. (Discussion section, page 17-18, line 278-289)

In addition,

This study revealed substantial disparities in IDSR implementation across healthcare tiers in Sidama Region. Training coverage declined sharply from district to peripheral levels, with only 11% of health posts and 12% of HDA members receiving relevant training, reflecting a clear cascade gap consistent with patterns reported elsewhere in East Africa. In Tanzania, Saleh et al. [32] documented a similar gradient, with training coverage declining from 78% at district level to 34% at dispensary level. Likewise, supervisory visits reached only 61% of health posts in Sidama, comparable to findings from Kenya, where Ng'etich et al. [30] reported that only 41% of health posts had received supervisory visits. Within Ethiopia, our findings align with Alemu et al. [34], who evaluated the Dangila district surveillance system and similarly identified training gaps (only 45% of health workers trained) and inconsistent supervision as major barriers. However, the present study extends these findings by quantifying the cascade gap specifically at the HDA level (4.2% case definition adherence), a community-tier actor not included in previous Ethiopian evaluations. In Iraq, Hamalaw et al. [35] found that more than 25% of outbreaks were not notified within mandated timeframes, consistent with the preparedness and reporting delays observed in Sidama. Taken together, these cross-country patterns suggest that the challenges identified are not unique to Ethiopia but reflect broader systemic weaknesses in decentralized surveillance systems across comparable low- and middle-income settings. (Discussion section, page 19-20, line 318-134)

Comment 1.6: Actionable Policy Recommendations

Comment: Providing more actionable policy recommendations would substantially improve the manuscript.

Response: We have added a dedicated Implications and Recommendations section with specific, tier-tailored, actionable recommendations.

Revisions: The following actionable recommendations emerge from the findings:

1. Establish dedicated outbreak response budgets at district level and ensure regular supply of surveillance forms at health posts.

2. Redesign training programs to reach peripheral levels, supported by simplified case definition job aids and quarterly supervisory visits.

3. Institutionalize routine data quality audits with feedback loops from regional to community levels, using predefined benchmarks as monitoring targets.

Strengthening IDSR requires tier-specific interventions that address cascaded weaknesses at peripheral levels rather than one-size-fits-all approaches. (Implications and Recommendations section, page 21, line 356-364)

Comment 1.7: Inconsistencies and Repetition

Comment: Some inconsistencies between figures reported in the text and tables, repetition across sections, and excessive length in certain parts should be addressed.

Response: We have carefully reviewed the manuscript and corrected all identified inconsistencies. Specifically:

1. Data Analysis percentages (Table 1): Corrected district-level analysis by person from 73% to 85% (11/13 facilities)

2. Feedback percentages (Table 1): Corrected public facilities feedback from 70% to 63% (17/27 facilities)

3. Removed duplicate paragraph justifying Sidama Region selection in Background section

4. Standardized terminology: "Health Development Army (HDA)" used consistently throughout

Response to Reviewer #2

General Comment: The reviewer states the manuscript is technically sound and methodologically appropriate, with strong alignment to WHO IDSR frameworks and inclusion of community-level actors as a major strength.

Response: We thank the reviewer for the thorough and constructive assessment. We have addressed each recommendation below.

Comment 2.1: Distinguish Structural Capacity Gaps from Performance Gaps

Comment: More explicitly distinguish structural capacity gaps from performance or outcome gaps.

Response: We have revised the Discussion to explicitly distinguish between structural capacity (availability of plans, budgets, personnel, guidelines) and operational performance (adherence to case definitions, data quality, trend analysis).

Revisions: The observed disparities likely reflect underlying structural and managerial factors rather than isolated individual performance gaps. District-level offices receive direct budget allocations for training and supervision from the regional health bureau, whereas health posts and HDA members rely on intermittent, often unfunded support from health centers. This cascaded model of capacity building, while efficient for centralized resource allocation, results in a gradual dilution of training quality and reduced frequency of supervision at lower tiers. (Discussion section, page 17-18, line 278-283)

Comment 2.2: Explicit Performance Thresholds to Avoid Overgeneralization

Comment: Avoid occasional overgeneralization (e.g., 'majority of sites performed inadequately') without defining explicit performance thresholds.

Response: We have removed all vague generalizations and replaced them with threshold-referenced statements throughout the manuscript.

Revisions: Only 11% of health posts and 12% of HDA members reported receiving relevant training. Thus, neither health posts nor HDA members met the predefined ≥80% training coverage benchmark. (Abstract section, lines 41-42)

Additionally, in result section,

Using the a priori performance thresholds defined in the Methods (≥90% for formal facilities, ≥80% for community actors), we evaluated adherence across all levels. (Results section, page 9, line 194-196)

Comment 2.3: Explicit Justification for No Inferential Statistics

Comment: Explicitly stating why inferential statistics were not pursued (e.g., non-probability sampling at later stages, system evaluation intent rather than hypothesis testing).

Response: We have added explicit justification as requested.

Revisions: Fourth, inferential statistics were not employed a deliberate choice given the descriptive, evaluative design, non-probability sampling, and hierarchical clustering of observations. Instead, we provide descriptive estimates with benchmarked interpretations, which are appropriate for systems evaluations. (Limitations and Strength section, page 20, line 341-344)

Comment 2.4: Confidence Intervals or Variability Measures

Comment: Reporting confidence intervals or variability measures for key proportions, where feasible.

Response: While we acknowledge this suggestion, we respectfully note that confidence intervals are typically associated with inferential statistical frameworks. Our study is explicitly descriptive and evaluative, designed to assess system performance against predefined benchmarks rather than to estimate population parameters or test hypotheses. The a priori thresholds (≥90% for formal facilities, ≥80% for community actors) provide the appropriate interpretive framework. Adding confidence intervals would imply a sampling-based inferential approach that does not align with our purposive sampling at lower tiers or our evaluative design. We have clarified this in the manuscript.

Revisions: Transparent reporting of performance benchmarks allows readers to interpret gaps without overreliance on statistical significance. (Limitations and Strength section, page 20-21, line 349-351)

Comment 2.5: Facility-Level Clustering Consideration

Comment: Clarifying whether facility-level clustering was considered when summarizing results across tiers.

Response: Thank you for this important methodological question.

We fully recognize that facility-level clustering exists in our data (e.g., multiple health posts within the same district, multiple HDA members within the same health post, and multiple facility types within the same catchment area).

However, for the following reasons, clustering was not statistically adjusted for in our analysis:

1. Descriptive and evaluative design: Our study was designed to describe system performance against predefined benchmarks, not to test inferential hypotheses. Clustering adjustments are typically required for inferential statistics (e.g., regression models with cluster-robust standard errors), which we did not perform.

2. Non-

---

## [Decision Letter · Decision Letter 1]

6 May 2026

Evaluating the Integrated Disease Surveillance and Response System in Sidama Region, Ethiopia: A Systems Evaluation

PONE-D-25-41162R1

Dear Dr. Sasie,

We’re pleased to inform you that your manuscript has been judged scientifically suitable for publication and will be formally accepted for publication once it meets all outstanding technical requirements.

Kind regards,

Shabnam Iezadi, Ph.D.

Academic Editor

PLOS One

Additional Editor Comments (optional):

Reviewers' comments:

Reviewer's Responses to Questions

**Comments to the Author**

1. If the authors have adequately addressed your comments raised in a previous round of review and you feel that this manuscript is now acceptable for publication, you may indicate that here to bypass the “Comments to the Author” section, enter your conflict of interest statement in the “Confidential to Editor” section, and submit your "Accept" recommendation.

Reviewer #2: All comments have been addressed

2. Is the manuscript technically sound, and do the data support the conclusions?

Reviewer #2: Yes

3. Has the statistical analysis been performed appropriately and rigorously? 

Reviewer #2: Yes

4. Have the authors made all data underlying the findings in their manuscript fully available?

Reviewer #2: Yes

5. Is the manuscript presented in an intelligible fashion and written in standard English?

Reviewer #2: Yes

6. Review Comments to the Author

Reviewer #2: Thanks to authors for adequately addressing all comments highlighted in a previous round of review. The manuscript is now acceptable for publication.

7. PLOS authors have the option to publish the peer review history of their article (what does this mean?). If published, this will include your full peer review and any attached files.

Reviewer #2: **Yes:** Mohammad J. Alhawajreh

---

## [Editor Report · Acceptance letter]

PONE-D-25-41162R1

PLOS One

Dear Dr. Sasie,

I'm pleased to inform you that your manuscript has been deemed suitable for publication in PLOS One. Congratulations! Your manuscript is now being handed over to our production team.

Kind regards,

on behalf of

Dr. Shabnam Iezadi

Academic Editor

PLOS One